# Formation of Nanostructure during Replication of a Hierarchical Plant Surface

**DOI:** 10.3390/nano11112811

**Published:** 2021-10-23

**Authors:** Dora Kroisová, Štěpánka Dvořáčková, Petr Kůsa

**Affiliations:** Faculty of Mechanical Engineering, Technical University of Liberec, 461 17 Liberec, Czech Republic; stepanka.dvorackova@tul.cz (Š.D.); petr.kusa@tul.cz (P.K.)

**Keywords:** plant surface, replication, hierarchical structures, nanostructure

## Abstract

Plant and animal surfaces have become a model for preparing special synthetic surfaces with low wettability, reflectivity, or antibacterial properties. Processes that lead to the creation of replicas of natural character use two-step imprinting methods. This article describes a technique of synthetic polymer surface preparation by the process of two-stage imprinting. The laboratory-prepared structure copies the original natural pattern at the micrometer and sub-micrometer levels, supplemented by a new substructure. The new substructure identified by the scanning electron microscope is created at the nanometer level during the technological process. The nanostructure is formed only under the conditions that a hierarchical structure forms the surface of the natural replicated pattern, the replication mold is from a soft elastomeric material, and the material for producing the synthetic surface is a polymer capable of crystallizing. A new nanometer substructure formation occurs when the polymer cools to standard laboratory temperature and atmospheric pressure.

## 1. Introduction

Plant and animal objects have become subjects of interest of research institutes in recent decades. They have become a model for a new perspective on such processes as the impact and use of sunlight, self-cleaning ability, antibacterial surface behavior, and more. The waxy nanostructured surfaces identified in many plants allow self-cleaning of leaf and flower surfaces, thus preventing a reduction in the efficiency of photosynthesis. Light-trapping plant surface structures minimize the reflection of incident light radiation [1,2,3,4,5]. The overall features of these surfaces are the presence of structures at the micrometer, sub-micrometer, and nanometer levels, and the existence of a sizeable specific character on which all-natural processes take place. The interest in imitating natural surfaces exhibiting these processes is related to the effort to translate these phenomena into technical practice and use them [6,7,8].

There are several possibilities to imitate natural textured surfaces. These include, for example, methods of imprinting—replication to various matrices, plating, laser, or etching [1,9,10,11]. From the point of view of the replication process, the appropriate choice of material for the preparation of the negative replica (mold) is essential, as well as the material for the production of the positive replica, the method of removing the replica from the mold, and possible separation of the mold. The applicability of metal molds is limited. It is challenging to produce a spatial structure in which micrometer, sub-micrometer, and nanometer dimensions will be interconnected. Removing the material from the mold can be complicated due to the large specific surface area. The cost of acquiring a metal mold with a hierarchical structure copying natural surfaces is high. A custom mold or pattern can be made with a focused ion beam. However, the method is very time- and cost-intensive [12,13,14]. Another way to create hierarchical structures is 3D printing. Although traditional 3D printers have the ability to print miniature objects, they cannot handle resolutions at the level of tens to hundreds of nanometers and especially the creation of hierarchical structures at this dimensional level. This printing would be possible using 3D nano- printers that use unique printing material. These methods are technologically demanding and often require more steps [15,16].

Commonly used laboratory processes, which lead to faithful replicas of natural surfaces, use two-stage imprinting methods. The natural surface serves as a pattern on which a layer of silicone material is applied. After removing the silicone mass from the natural character, a mold is created. The so-called first or negative replica is used to prepare the second positive replica, which is a copy of the natural surface. Epoxy resins or silicone materials that replicate the original textures are usually used to prepare images of natural characters. These methods make it straightforward to obtain identical polymer replicas that faithfully copy the natural pattern at the micrometer and sub-micrometer levels [8,13].

The experiment aimed to create replicas of hierarchical structures from semi-crystalline thermoplastics. The reason that led to the design of this experiment was that semicrystalline polymers such as polyethylene and polypropylene are inexpensive materials with a wide range of applications and are easy to process. The potential of these structured surfaces lies primarily in their large specific surface area. Applicability can be expected, especially in light-trapping structures and self-cleaning surfaces and unique surfaces with advanced thermal conductivity [17].

## 2. Materials and Methods

### 2.1. Plant Materials

The garden pansy (*Viola x wittrockiana*) was selected for the distinct hierarchical surface of crown leaves that occur in this plant. Thanks to this surface, the flowers show specific optical properties—a velvety appearance and, at the same time, superhydrophobic behavior.

Crown leaves (Figure 1) were used for the experiments, from which it was possible to cut out planar samples from 1 cm^2^ to 4 cm^2^. The thickness of the crown leaf samples ranged up to 100 μm in the state after collection. Because the plants contain up to 90% water, all surface replication experiments were performed immediately after sampling to avoid sample drying and thus undesired destruction of the crown leaf surface structure, as seen in the Figure 2. Before sampling the plant, the crown leaves were washed with distilled water to remove impurities—dust particles or pollen grains. The plants were left briefly in the air to dry any residual water on the surface.

### 2.2. Replication Procedure

A fresh non-dried sample of the crown leaf was attached with double-sided adhesive tape to the flat plastic dish surface so that the replicated surface was placed upwards. A few millimeters thick silicone elastomer (President Light Body, Coltene Whaledent) was evenly applied to the sample surface. The silicone mass was prepared by mixing the two components in a ratio of 1:1; the mixing time was not longer than 60 s, all according to the manufacturer’s instructions. The surface was load by a force of up to 5 N to facilitate the elastomer filling into the microscopic/submicroscopic structures of the biological sample. A load was applied throughout the crosslinking of the silicone elastomer. After crosslinking the elastomer for a maximum of 5 min, the elastomer layer was removed from the surface of the plant sample (Figure 3). Any plant residues adhering to the mold surface were removed manually using distilled water or chloroform (Sigma Aldrich Prague, Czech Republic)). The formed elastomeric layer—the so-called first (negative) replica, was used as an elastomeric mold. The thickness of this mold was up to 5 mm. Two crystallizing polymers were used to prepare polymer replicas—polyethylene (Sigma Aldrich, melting point 116 °C) and polypropylene (Sigma Aldrich, melting point 157 °C). The polymer in the form of powder, granules, or foil was placed in this mold and melted, creating a continuous film (Figure 3). The elastomer mold surface was loaded with a force of up to 10 N to ensure the required quality of polymer replica. After uniform melting of the polymer in the elastomeric mold, which took place for several minutes, the mold, together with the molten polymer, was removed from the oven and cooled to room temperature and atmospheric pressure. Upon cooling, the polymer crystallized in elastomeric mold, and a positive polymeric replica of the plant surface was formed [19].

### 2.3. Surface Characterization

Scanning electron microscopy was used to characterize the morphology of the surface structures of the plant pattern, the elastomeric mold, and the polymeric replicas. Samples of crown leaves, samples of elastomeric mold, and polymer replicas were excised from the prepared samples (Figure 4). To study the natural surface structure of the crown leaves of the garden pansy, a thin top layer removed from the flower was used, which was air-dried before microscopy (Figure 5). All evaluated models were attached to the aluminum target with double-sided carbon adhesive tape and were sputtered with a layer of Pt-Pd (Quorum Q150R ES) with a thickness of 2–4 nm. Characterization of all surfaces was performed on scanning electron microscopes ZEISS ULTRA PLUS and VEGA TESCAN.

## 3. Results

Figure 5a shows the top layer of surface cells located on the crown leaf of the garden pansy (*Viola x wittrockiana*). This layer contains a minimal amount of water, and thus the deformation of the conical formations during free drying in the air is negligible. In terms of morphology, the adaxial layer of the crown leaf of the garden pansy is formed by tightly arranged conical cells with dimensions at the hexagonal base of 30 to 40 µm; the height of the cells is about 40 µm. The conical cells have smooth folds on their surface (Figure 5b); the thickness and height range from 300 to 400 nm. These protrusions do not show any other substructure on their surface.

Microscopic images of the elastomeric mold—negative replicas (Figure 6a) show a quality replication process. The mold consists of conical holes formed when the elastomer was applied to the surface of a fresh plant pattern. It is an observable protrusion of sub-micrometer dimension on the hole walls, copying the plant cell surface. The arrangement of individual surface cones, their connection, and their shape are captured. A detailed microscopic image shows that the surface of the mold protrusions is smooth (Figure 6b).

Figure 7 shows a microscopic image of a positive replica formed by melting and subsequent crystallization of the polymer in a soft elastomeric mold. As can be seen from the picture, the ability of the polymer to fill the elastomeric mold was excellent. Conical cells corresponded in their shapes, dimensions, arrangement, and morphology to the hierarchical structure of plant pattern cells. The polymer flowed precisely into the mold and filled it.

A detailed microscopic image (Figure 8) shows the excellent structure found on the surface of these protrusions. The structure is made up of “fine fibers” measuring tens of nanometers. These fibers cover both the surface of the protrusions and the depressions between the protrusions. From a comparison with the above images of the biological sample (Figure 5) and the images of the elastomeric mold (Figure 6), it is clear that this structure is new and was not visible on the original plant sample or the mold surface.

## 4. Discussion

The nanostructure documented by an electron microscope was formed by cooling a molten crystallizing polymer in a soft elastomeric form with a hierarchical surface at atmospheric pressure and room temperature. Plastics processors use pressures and repressures (on the order of up to 150 MPa) in their technological processes, which enable perfect filling of the mold and eliminate product defects. The surfaces of the molds are smooth so that there are no problems when removing the manufactured parts. These standard processes use a combination of a smooth surface of a cooled metal mold and pressure in the mold during cooling. The process creates a skinny layer on the surface of the standard product, influenced by the flow direction of the melt, in which the crystallization process is limited.

In contrast, the above-described process for preparing a hierarchical structure operates with a soft elastomeric form in which the crystallizing polymer cools at atmospheric pressure. The cooling temperature decreases from approx. 150–180 °C, depending on the type of polymer used, to a laboratory temperature of approx. 22 °C. Due to the small thickness of the sample (1–2 mm), cooling took place for several minutes. The evaluation of all models took place 14 days after their preparation to eliminate possible changes over time in the structure associated with primary and secondary crystallization processes.

The soft elastomeric form allowed the crystallizing polymer to form a fibrous structure consisting of fibers with dimensions of 10 to 15 nm, which were analogous to the so-called lamellae formed during the crystallization of polymers [21]. The presence of the folds/protrusions described above with a thickness and height of 300 to 400 nm is essential for forming the fibrous structure. As verified, without the presence of these folds, no fibrous structure is formed. The fibers have homogeneous dimensions. Accessible spaces are created between the threads with measurements at units up to tens of nanometers. The resulting hierarchical structure is not affected by the cooling rate of the samples, the polymer melting load, or the thickness of the models. Polymers with better crystallization ability show a more ordered structure [20].

The scanning electron microscope was the most suitable technique for the complex characterization of the created hierarchical structure. The formation of a nanostructure not on the original natural pattern is conditioned by the simultaneous combination of all the above conditions.

The soft elastomeric mold of the above material is reusable, ensuring quality transfer of the hierarchical surface structure and the formation of a new substructure. A limiting factor in the use of the mold is the higher temperatures, which lead to its degradation after a more extended period of use.

The importance of the hierarchical polymer structure lies mainly in the potential possibilities for its use. The process of two-stage replication created designs that were tested for wettability and self-cleaning ability, respectively (Figure 4) [20]. Initial experiments with the behavior of aqueous solutions were performed on the hierarchical structure of the prepared cup-shaped bodies (Figure 9). They led mainly to the issue of crystallization on nanostructured surfaces.

## 5. Conclusions

A simple procedure created a new substructure during the two-stage replication of the hierarchical plant surface. This nanostructure was not on the original natural pattern. This substructure was built up of fibers with dimensions in tens of nanometers. Its origin was primarily conditioned by the presence of a hierarchical structure on the plant pattern. Secondly, it was conditioned by the simultaneous use of soft elastomeric material to produce a mold, a crystallizing polymer for replicas, and the technological conditions for the process—cooling of the polymer at room temperature and atmospheric pressure.

The created hierarchically structured surface has considerable potential in further experiments dealing with the issue of the impact and reflection of solar radiation, the so-called light-trapping structures, aimed at increasing the efficiency of solar power systems. Accessible technology for preparing designs imitating non-wetting plant surfaces provides opportunities for further research and possible modification of these structures to create functional self-cleaning surfaces for technical applications. A highly structured surface was also presented to study the adsorption of liquid substances and the kinetics of the processes taking place on said nano-surfaces. The main advantage of the hierarchical structures described above is the simplicity and feasibility of implementing the process without expensive laboratory equipment.

## Figures and Tables

**Figure 1 nanomaterials-11-02811-f001:**
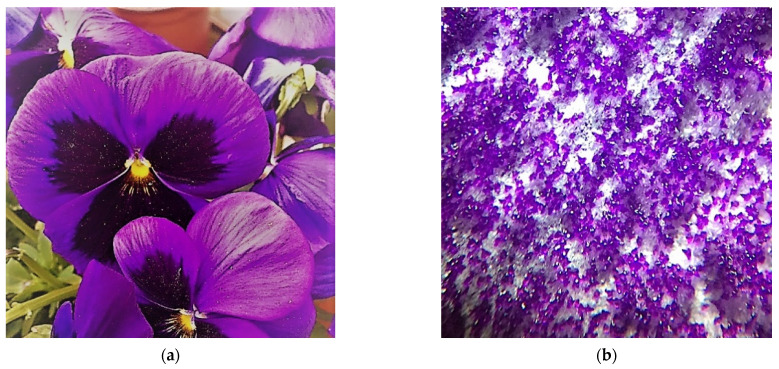
Garden pansy (*Viola x wittrockiana*). (**a**) Photo of the crown petals of a garden pansy flower. (**b**) Image of the flower petal surface taken from a 3D microscope Zeiss Stemi DV4, magnification 32 times.

**Figure 2 nanomaterials-11-02811-f002:**
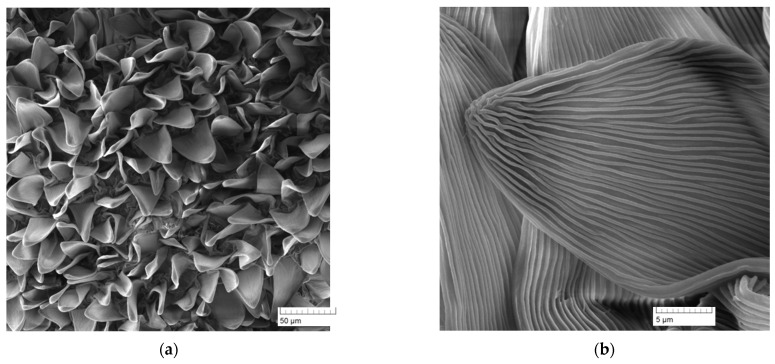
Garden pansy (*Viola x wittrockiana).* (**a**) Overview image of the dried crown leaf surface. (**b**) Detailed image of the individual cone surface with visible sub-micrometer folds. Reprinted from ref. [18].

**Figure 3 nanomaterials-11-02811-f003:**
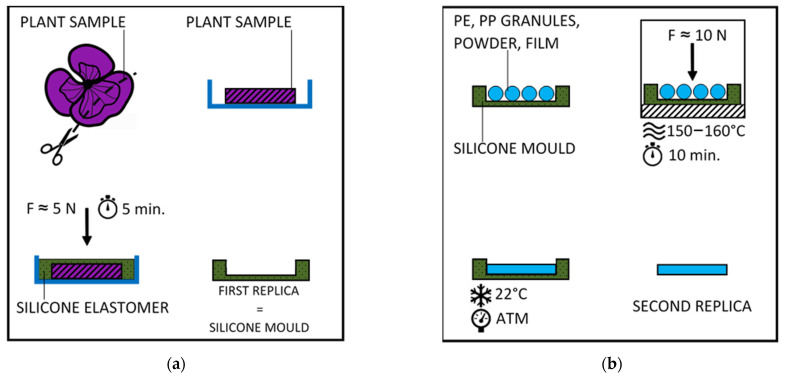
Replication process. (**a**) The method of forming the first (negative) elastomeric replica. (**b**) The method of forming the second (positive) polymer replica.

**Figure 4 nanomaterials-11-02811-f004:**
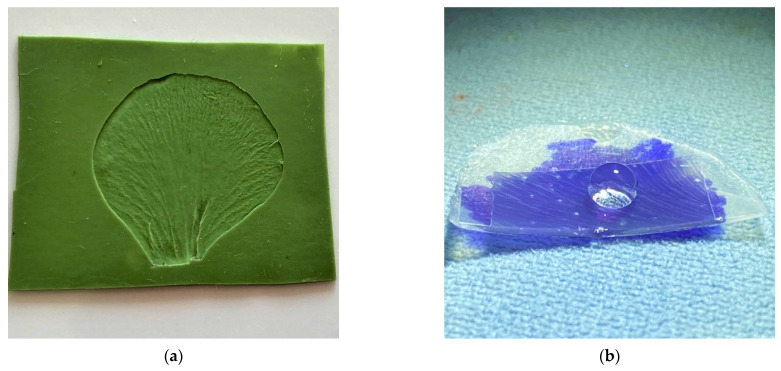
Replication results. (**a**) First (negative) replica—elastomeric mold. (**b**) Second (positive) replica—sample made of semi–crystalline polymer—drop of water on the surface demonstrating the hydrophobicity of the surface.

**Figure 5 nanomaterials-11-02811-f005:**
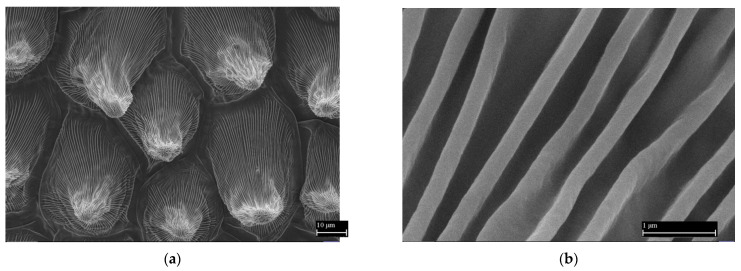
The surface of the crown leaves of the garden pansy (*Viola x wittrockiana*). (**a**) Overview microscopic image of a thin layer taken from the surface of the garden pansy crown leaf. (**b**) Detailed image of the surface of this layer with smooth sub-micrometer folds/protrusions. Reprinted from ref. [14,19,20].

**Figure 6 nanomaterials-11-02811-f006:**
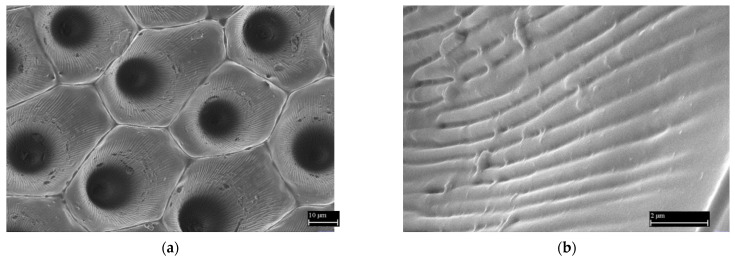
The surface of the elastomeric mold. (**a**) Overview microscopic image of the elastomeric mold—the first (negative) replica. (**b**) Detailed image of the mold surface. Reprinted from ref. [20].

**Figure 7 nanomaterials-11-02811-f007:**
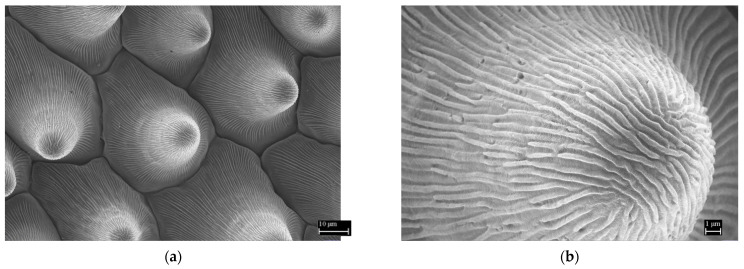
Polymer replica surface. (**a**) Overview microscopic image of the garden pansy polymer replica surface made of a crystallizing polymer. (**b**) Detailed image of the protrusions on the replica conical structure. Reprinted from ref. [19,20].

**Figure 8 nanomaterials-11-02811-f008:**
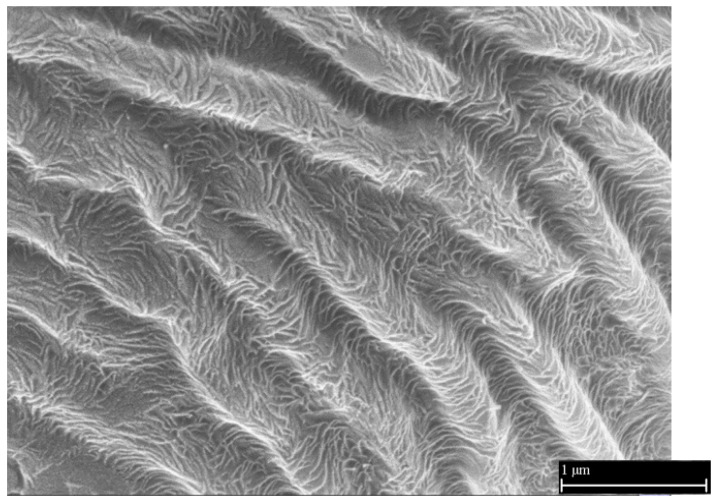
Polymer replica surface. Detailed microscopic image of the polymer replica surface made of a crystallizing polymer with visible nanostructure. Reprinted from ref. [19].

**Figure 9 nanomaterials-11-02811-f009:**
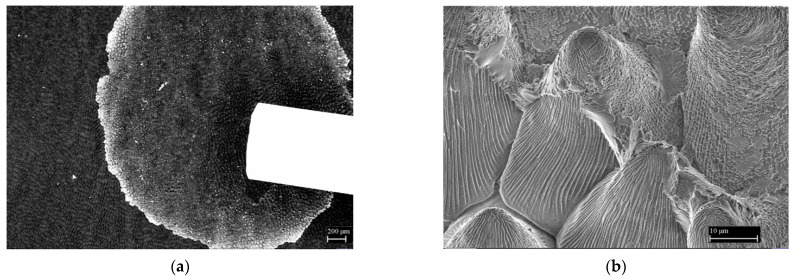
Polymer replica surface. (**a**) A drop of drinking water dries on the polymer surface of the replica—the white rectangle is part of SEM blowing nitrogen on the surface. (**b**) Detailed image of the interface between the water-wetted and non-wetted parts of the polymer surface.

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
