# Peer review of "Formation of Nanostructure during Replication of a Hierarchical Plant Surface"

_nanomaterials, 2021, doi:10.3390/nano11112811_

Round 1
Reviewer 1 Report
The paper aims to investigate Formation of Nanostructure During Replication of a Hierar-2 chical Plant Surface. It describes a technique of synthetic polymer surface preparation by the process of two-stage imprinting. A new nanometer substructure formation occurs when the polymer cools to standard laboratory temperature and atmospheric pressure. Overall, it meets the high quality of nanomaterials, and my recommendation is to publish this paper with some minor revisions given below.
- Although the work of this manuscript has valuable contribution to the science and practical application, the author should emphasize the significance of the work in the manuscript.
- Refine the article, concisely and accurately elaborate the content of the manuscript, reduce redundant expressions.
- Some of the latest references are recommended reference in Introduction. For example:
- Junhui Li, Xiang Li, Yu Zheng, Zhan Liu, Qing Tian, Xiaohe Liu, New underfill material based on copper nanoparticles coated with silica for high thermally conductive and electrically insulating epoxy composites, Journal of Materials Science, 2019, 54(8), 6258-6271
- Liu, J. Li, X. Liu. (2020) Novel functionalized BN nanosheets/epoxy composites with advanced thermal conductivity and mechanical properties,ACS Applied Materials & Interfaces, 12(5), 6503-6515
- Li, X. Zhang, C. Zhou, J. Zheng, D. Ge, and W. Zhu, “New applications of an automated system for high-power LEDs,” IEEE-ASME Trans. Mech., 2016, 21(2):1035-1042.
Author Response
Dear Reviewer,
Based on your suggestions, I emphasized the importance of the work - see Introduction - last paragraph and Conclusion. According to your recommendation, I have selected and supplemented one of the references you have proposed to correspond to the focus of the problem. I increased the Discussion to clarify the text and reduce redundant expressions.
Regards
Dora Kroisová
Reviewer 2 Report
The objective of the work consists in the description of a simple and low cost procedure used for the creation of a new substructure during the two-stage replication of the hierarchical plant surface, obtained by pansy leaves.
As stated by the authors, this work could lay the groundwork for the development of natural-based model materials with large specific surface area.
In my opinion, the quality of the manuscript is high in terms of perspectives (even if not properly described), average in terms of characterizations performed, process explanations and readibility.
The major issues regard:
- the absence of future applications that must be reported in the conclusions,
- the lack of a better characterization of the new sub-structure obtained (nanopattern shape and size dependance respect to temperature and pressure variation)
- a discussion of what reported in Fig. 9 in terms of hydrophilicity and experimental procedure. Then, the figure itself is not clear (especially Fig. 9a, where the role of the white rectangular shape should be described and in my opinion should be replaced).
Here some further (minor) comments/suggestions:
1) Line 111: in the caption of Fig. 4b, the presence of the water drop should be reported and further discussed in the manuscript
2) Line 140: the phrase “you can see” in my opinion is not appropriate for a scientific manuscript. It could be replaced with “It is observable”.
3) Line 151: the term “nicely” could be replaced with “precisely”
Author Response
Dear Reviewer,
Based on your suggestions, I added potential applications - see Conclusion. I reworked the discussion to characterize better the processes of creating a new structure and the structure itself. I added a description to Figure 9 and explained it in the text - see the discussion,
I added the caption in Figure 4, revised the phrase on line 140 and the expression on line 151 of the original document.
Regards
Dora Kroisová
Round 2
Reviewer 2 Report
The authors properly replied to my comments. In my opinion, manuscript is now ready for being published in Nanomaterials.